# How Antidepressant Drugs Affect the Antielectroshock Action of Antiseizure Drugs in Mice: A Critical Review

**DOI:** 10.3390/ijms22052521

**Published:** 2021-03-03

**Authors:** Kinga K. Borowicz-Reutt

**Affiliations:** Independent Unit of Experimental Neuropathophysiology, Medical University of Lublin, Jaczewskiego 8b, PL-20-090 Lublin, Poland; kingaborowicz@umlub.pl

**Keywords:** antidepressant drugs, antiepileptic drugs, interactions, maximal electroshock

## Abstract

Depression coexists with epilepsy, worsening its course. Treatment of the two diseases enables the possibility of interactions between antidepressant and antiepileptic drugs. The aim of this review was to analyze such interactions in one animal seizure model—the maximal electroshock (MES) in mice. Although numerous antidepressants showed an anticonvulsant action, mianserin exhibited a proconvulsant effect against electroconvulsions. In most cases, antidepressants potentiated or remained ineffective in relation to the antielectroshock action of classical antiepileptic drugs. However, mianserin and trazodone reduced the action of valproate, phenytoin, and carbamazepine against the MES test. Antiseizure drug effects were potentiated by all groups of antidepressants independently of their mechanisms of action. Therefore, other factors, including brain-derived neurotrophic factor (BDNF) and glial-derived neurotrophic factor (GDNF) modulation, should be considered as the background for the effect of drug combinations.

## 1. Introduction

It is estimated that depression affects 350 million people worldwide. According to the World Health Organization data, it will be the most common disease in the world by 2030. Currently, depression takes a million lives a year from suicide. Depressive disorders are diagnosed in 2% of children, 8% of teenagers, and 10% of pregnant women. It is worth emphasizing that the percentage of pharmacological antidepressant treatment is also continuously increasing in these groups of patients [1]. On the other hand, depression and related disorders are the most common comorbidities in epilepsy patients. The coexistence of these disease entities reaches 30% and depressive episodes have begun to be considered a marker of drug-resistant epilepsy [2].

Evidently, depression in epilepsy patients requires appropriate treatment, including pharmacotherapy. Antidepressant drugs are used not only in the treatment of endogenous depression and depression accompanying neurodegenerative diseases, but also in that of phobias, compulsive-obsessive disorders, insomnia (a frequent mask of depression), fibromyalgia, and even obesity. The use of anticonvulsants also goes well beyond epilepsy and includes seizures accompanying other disturbances of the central nervous system (tumors, ischemia, and trauma), neuralgias, neuropathies, or eclampsia [2,3]. Such a wide use markedly increases the likelihood of interactions occurring between antiepileptic and antidepressant drugs.

Neurons in the brain of epilepsy patients are characterized by an increased excitability that may constitute a genetic background of this disorder. Therefore, every factor that increases the neuronal excitability may contribute to increases in the seizure frequency and severity. Such factors are, for example, a markedly increased body temperature (febrile seizures and heat shock), hypoglycemia, or drugs affecting the neuronal membrane potential, including antidepressant drugs. Currently, it is widely recognized that most antidepressant drugs are entirely safe in epilepsy patients at therapeutic doses. The confirmed exceptions are bupropion, clomipramine, amoxapine, and maprotiline [4,5]. A great deal of experimental research has shown the anticonvulsant effects of antidepressant drugs. Even more importantly, it was reported that norepinephrine reuptake inhibitors (reboxetine and atomoxetine), serotonin reuptake inhibitors (fluoxetine and citalopram), and the dual serotonin/norepinephrine reuptake inhibitor (duloxetine) reduced the incidence of seizure-induced respiratory arrest and death in mice subjected to the maximal electroshock test, which may suggest their potential protective effect against sudden unexpected death in epilepsy (SUDEP) [6].

The purpose of this review was to answer the question of whether antidepressant drugs can change the course of epilepsy and/or the course of antiseizure treatment. The literature on the effects of antidepressant drugs on seizures seems to be inconsistent. Either pro- or anticonvulsant effects, depending on the dose, animal species, seizure model, and administration protocol, have been described. Therefore, this study focused on experiments conducted on the maximal electroshock in mice (MES), representing the most used screening model for potential antiepileptic drugs active against generalized tonic-clonic convulsions. In this model, the action of antidepressant drugs on the electroconvulsive threshold and the anti-electroshock effect of antiseizure drugs have been evaluated.

## 2. Selection of Literature

The review is mainly based on the author’s own research. Medline and Science Direct databases were searched for additional articles using the term “antidepressant drugs and maximal electroshock”. Out of 71 (Medline) and 157 (Science Direct) results, 24 articles were selected as the most relevant.

## 3. Antidepressants Have Different Effects on Tonic-Clonic Convulsions

### 3.1. Tricyclic and Tetracyclic Antidepressant Drugs

Tricyclic drugs of the old generation (e.g., amitriptyline, imipramine, and desipramine) block the reuptake of serotonin and norepinephrine from the synaptic cleft. Maprotiline, which is a tetracyclic antidepressant, is an inhibitor of norepinephrine reuptake. However, adverse effects mainly related to their anticholinergic properties limited the use of the two types of drug in psychiatric practice.

In experimental conditions, amitriptyline (20–30 mg/kg, i.p.) and imipramine (30–40 mg/kg) exhibited their own anticonvulsant effects, significantly increasing the threshold for electroconvulsions. Desipramine (up to 40 mg/kg) remained ineffective in this aspect. The three tricyclic antidepressants, at doses subeffective in the electroconvulsive test (10 and 20 mg/kg, respectively), distinctly potentiated the protective efficacy of valproate against the MES. Pharmacokinetic data are incomplete. It is only known that desipramine (20 mg/kg) did not affect the brain concentration of valproate [7]. Furthermore, in another study, imipramine (17.5–25 mg/kg) showed an anticonvulsant action against the MES in mice. However, higher doses induced a neurotoxic effect, including clonic seizures [8]. The antielectroshock effect of combinations of amitriptyline and phenobarbital or carbamazepine was reported by Tregubov and Kolla [9]. Doxepin was effective against maximal electroshock-induced seizures, with an estimated 50% effective dose (ED_50_) value of 6.6 mg/kg [10].

### 3.2. Selective Serotonin Reuptake Inhibitors

Acute and chronic injections of fluoxetine (10 mg/kg) did not modify the threshold for electroconvulsions in mice [11]. On the other hand, this antidepressant, when applied once with the dose range of 15–25 mg/kg, significantly increased the threshold [12]. In contrast, chronic (14-day) treatment with fluoxetine (up to 20 mg/kg) did not affect this parameter [13]. A recent publication revealed that raloxifene, which is a selective modulator of estrogen receptors and a potential antiviral medication in the treatment of coronavirus disease 2019 (COVID-19), enhanced the action of fluoxetine (14 mg/kg) in the MES test [14]. Fluoxetine, when applied once at subeffective doses against the electroconvulsive threshold, significantly enhanced the anticonvulsive activity of carbamazepine, phenobarbital, and phenytoin, but not valproate, against the MES in mice [12,15]. The chronic administration of this antidepressant (15–20 mg/kg) led to an enhancement of the protective activity of all four classic antiepileptic drugs [13]. In another study, fluoxetine, given either acutely or chronically at the lower dose of 10 mg/kg, did not affect the action of phenytoin in the MES test [11]. A pharmacokinetic verification of interactions indicated that acute administration of the antidepressant significantly increased brain levels of carbamazepine and phenobarbital, while chronic treatment elevated brain concentrations of all four antiepileptic drugs [12,13].

For other selective serotonin reuptake inhibitors, the only data are that paroxetine did not affect the MES-induced convulsions in mice [16].

### 3.3. Serotonin and Noradrenaline Reuptake Inhibitors

Both acute and chronic venlafaxine (12.5–25 mg/kg) raised the electroconvulsive threshold in mice. At subprotective doses, this antidepressant enhanced the action of valproate against MES-induced seizures in mice. At the lowest effective doses, venlafaxine additionally intensified the action of carbamazepine and phenobarbital. However, the interaction with phenobarbital could be partially due to pharmacokinetic events, since venlafaxine increased the brain concentration of phenobarbital. In the same study, chronic (14-day) treatment with venlafaxine enhanced the action of phenytoin, despite decreasing the brain level of this antiepileptic drug [17].

Milnacipran (10 mg/kg), when given once, but not repeatedly, raised the electroconvulsive threshold in mice. Furthermore, acute but not chronic treatment with this antidepressant potentiated the action of four first-generation antiepileptic drugs against the MES test in mice. The anti-electroshock effects of carbamazepine and phenobarbital were enhanced by subthreshold doses of milnacipran. Higher doses of the antidepressant were required to affect the action of valproate and phenytoin. Interestingly, no changes in brain concentrations of antiepileptic drugs were observed, so the contribution of pharmacokinetic events to revealed interactions is rather unlikely [18].

Duloxetine exhibited an anticonvulsant action, with the estimated ED_50_ value of 48.21 mg/kg in the MES test in mice. The single administration of duloxetine (6.25–25 mg/kg) significantly increased the anti-electroshock effects of valproate, carbamazepine, and oxcarbazepine. Pretreatment with an inhibitor of gamma-aminobutyric acid (GABA) synthesis significantly increased the ED_50_ of duloxetine, suggesting the role of the GABAergic system in the mechanism of duloxetine’s action [19].

### 3.4. Norepinephrine Reuptake Inhibitors

An increase in the electroconvulsive threshold was observed when reboxetine (8–16 mg/kg) was applied only once, not repeatedly. In a continuation of this study, the influence of this antidepressant on the antielectroshock action of chosen antiepileptic drugs was evaluated. A single injection of reboxetine at subthreshold doses enhanced the effectiveness of carbamazepine, valproate, and phenobarbital. At higher doses, the antidepressant drug potentiated the action of phenytoin. The chronic (14-day) administration of reboxetine only enhanced the action of carbamazepine [20].

### 3.5. Norepinephrine and Dopamine Inhibitors

The acute administration of bupropion led to a biphasic effect in the MES test in mice. On the one hand, this antidepressant demonstrated clear antielectroshock activity, with an ED_50_ value of 19.4 mg/kg. At higher doses, bupropion was proconvulsant, with a 97% convulsive dose (CD_97_) value of 139.5 mg/kg [21,22]. When the antidepressant was administered chronically (5 mg/kg), it increased the threshold for electroconvulsions and enhanced the antielectroshock action of felbamate, lamotrigine, and topiramate. The interaction with lamotrigine could be at least partially of a pharmacokinetic nature. However, intensification of the effect of topiramate occurred, despite decreased brain concentrations of this antiepileptic [23].

### 3.6. Selective Reversible Monoamine Oxidase (MAO) Inhibitors

Given in a single injection, moclobemide (62.5 and 75 mg/kg) increased the electroconvulsive threshold. In contrast, chronic treatment with this antidepressant did not influence the threshold. Acute moclobemide applied at subthreshold doses (up to 50 mg/kg) enhanced the antielectroshock effects of carbamazepine, valproate, and phenobarbital. Chronic moclobemide (37.5–75 mg/kg) increased the action of all four antiepileptic drugs. All revealed interactions, except for those between moclobemide and phenobarbital, seem to have a pharmacokinetic nature, because the antidepressant drug, either in acute or chronic treatment, increased the brain concentrations of respective antiepileptic drugs [24].

### 3.7. Other Antidepressant Drugs

Mianserin increases serotonergic and noradrenergic neurotransmission by acting as an antagonist at the serotonin 5-HT_2_ and adrenergic α_2_ presynaptic and somatodendritic auto- and heteroreceptors. Acute mianserin (30–40 mg/kg) increased the electroconvulsive threshold in mice, while chronic treatment led to the opposite effect. Moreover, a single application of this antidepressant at subthreshold doses enhanced the antielectroshock action of carbamazepine, valproate, and phenytoin, but not that of phenobarbital. In contrast, chronic (14-day) treatment reduced the effect of valproate and phenytoin against maximal electroshock. All revealed interactions seemed to have a pharmacodynamic nature [25].

Trazodone is an antidepressant inhibiting serotonin transport and blocking 5HT_2_ receptors. Chronic (but not acute) treatment with trazodone (10–40 mg/kg) increased the threshold for electroconvulsions, exhibiting a clear antiseizure action. However, when applied acutely or chronically (at its subthreshold doses), this antidepressant diminished the antielectroshock activity of carbamazepine and phenytoin. The effects of valproate and phenobarbital remained unchanged. Pharmacokinetic interactions between trazodone and the antiepileptic drugs were not observed [26].

Tianeptine is an antidepressant drug that contradicts the monoaminergic theory of depression. This medication even enhances serotonin uptake from the synaptic cleft. Tianeptine seems to act as an agonist of opioid receptors and mediate a variety of neurobiological effects, including the restoration of plasticity in the amygdala, decrease in stress-induced glutamate release, and reversal of stress-induced dystrophy of hippocampal CA3 dendrites [27]. The antiseizure effect of tianeptine can also be explained by purinergic (through adenosine A1 receptors), glutamatergic, and nitrergic mechanisms [28]. Neither acute nor repeated treatment with tianeptine (up to 50 mg/kg) affected the electroconvulsive threshold in mice. Nevertheless, both acute and chronic application potentiated the antielectroshock action of carbamazepine, valproate, and phenobarbital, but not phenytoin. The action of phenobarbital was even enhanced when lowering its brain level. Brain concentrations of remaining antiepileptics were not affected by tianeptine [28].

All data can be found in Table 1 and Table 2.

### 3.8. Newer Drugs with Antidepressant Activity

There are currently no data on the effects of drugs such as esketamine, allopregnanolone (brexanolone), vortioxetine, or agomelatine in the *MES* test. Interactions between the above-mentioned drugs and antiepileptic drugs have also not been studied in this seizure model. However, allopregnanolone interacted synergistically with tiagabine in two other models of electricaly-induced convulsions, including hippocampus kindling and 6-Hz seizure models in mice [29].

However, ketamine, which is the parent drug for esketamine mostly used as an injectable anesthetic, was investigated in the MES test. Ketamine exhibited the characteristic of an antiepileptic drug in this test and synergistically interacted with valproate and carbamazepine. Combinations of ketamine and phenytoin or phenytoin led to additivity. The determined interactions do not appear to be dependent on pharmacokinetic events, since none of the injectable anesthetics changed the free plasma levels of respective antiepileptic drugs [30].

## 4. Discussion

### 4.1. Probable Mechanisms of the Anticonvulsant Action of Antidepressant Drugs

The most obvious mechanism used to be associated with enhanced serotonergic neurotransmission. Several studies proved that animals with genetically conditioned seizures exhibited serotonin and/or noradrenergic deficits, while enhanced serotonergic and/or noradrenergic transmission attenuated seizures [31]. However, the issue of serotonergic’s action seems to be more complicated. For instance, tool substances HBK-14 and HBK-15, which are two triple 5-HT_1A_, 5-HT_7_, and 5-HT_3_ antagonists (HBK-14 and HBK-15), were reported to have potent antidepressant-like and anticonvulsant properties in a variety of animal models [32]. On the other hand, mirtazapine activating serotonergic 5-HT_1A_ receptors and blocking 5-HT_2_ and 5-HT_3_ receptors did not affect seizures in the maximal electroshock [33]. The results of the two studies considered together may suggest that the anticonvulsant action could be mainly related to blocking of the 5-HT_7_ receptor.

The probable anticonvulsant effect of dopamine is believed to be realized through activation of the D_2_ receptor, while that of noradrenaline is believed to occur through the stimulation of alpha_2_- and beta_2_-adrenoreceptors. Acetylcholine may play a dual role in the pathogenesis of experimental seizures. Pilocarpine, which is a muscarinic agonist, induces primary limbic seizures with secondary generalization. However, another muscarinic agonist, called carbachol, reduced epileptiform discharges in the model of absence seizures [31].

On the other hand, mechanisms conditioning the anticonvulsant action, such as the blocking of sodium or calcium channels and activation of potassium channels, may contribute to antidepressant-like behavior in experimental animals. Moreover, ketamine, which is an antagonist of glutamatergic N-methyl-D-aspartate (NMDA) receptors, exhibits not only anticonvulsant, but also antidepressant-like, effects in animal models [31]. The antielectroshock action of ketamine may also be associated with its action on D_2_/sigma, GABA_A_, and GABA_B_ [34].

There are some reports confirming the influence of antidepressant drugs on glutamate release in different brain regions. For instance, fluoxetine decreased glutamate synaptic release. This effect can be explained by changes in the structure of the soluble *N*-ethylmaleimide-sensitive factor attachment protein receptor (SNARE) complex, the effector of vesicle docking, and fusion at presynaptic membranes [35]. The same mechanism seems to be involved in the action of chronically administered venlafaxine. This drug was reported to dampen glutamate release from rat hippocampal synaptic terminals and the frontal/prefrontal cortex [36]. Chronic (but not acute) treatment with fluoxetine, reboxetine, and desipramine reduced the depolarization-evoked release of glutamate in the rat hippocampus [37]. In turn, milnacipran reduced the amplitude of NMDA-, but not α-amino-3-hydroxy-5-methyl-4-isoxazolepropionic acid (AMPA)-mediated currents in lamina II neurons of rat spinal cord slices [38] and blocked activated NMDA receptors in Xenopus oocytes [39]. Bupropion inhibited the 4-aminopyridine-evoked release of glutamate from rat cerebrocortical nerve terminals [40,41]. Moreover, bupropion increased the seizure latency and decreased the seizure severity in the kainic acid rat seizure model, representing an animal model for temporal lobe epilepsy [38]. Trazodone, as a partial agonist of presynaptic 5-HT_2A_ heteroreceptors, decreased the release of glutamate from rat cerebellar mossy fibers [42]. Finally, tianeptine modulates glutamatergic transmission through the activation of mu opioid receptors [43].

On the other hand, the activity of antidepressants with a serotonergic mechanism of action (escitalopram and milnacipran) can be inhibited by NMDA agonists, while antidepressants with a noradrenergic mechanism of action (imipramine and reboxetine) can be inhibited by AMPA antagonists [44]. This may be indirect proof that antidepressant drugs diminish glutamatergic transmission.

Similarly, seizure inhibition and the antidepressant-like action may be mediated by the enhancement of GABA-ergic neurotransmission. In contrast, GABA deficits were reported to inhibit serotonergic and noradrenergic neurotransmission in the brain. Moreover, the dual anticonvulsant/antidepressant effect can be attributed to the action of some neurotrophins (brain-derived neurotrophic factor (BDNF) and glial-derived neurotrophic factor (GDNF)) and neurosteroids (allopregnanolone) [31].

A plethora of data indicate that neurotrophic factors, above all the brain-derived neurotrophic factor (BDNF) and glial-derived neurotrophic factor (GDNF), participate in processes of brain plasticity and epileptogenesis. Interestingly, BDNF shows excitatory effects in in vitro conditions, neuronal cultures, and animal brain slices. Furthermore, the acute intracerebral administration of BDNF induces seizures in mice. In contrast, the chronic infusion of BDNF reduces the neuronal excitability, which can be related to the downregulation of tropomyosin receptor kinase B (TrkB, the receptor for BDNF) and, in consequence, increased brain concentrations of neuropeptide Y (NPY) and altered chloride conductance. Furthermore, NPY proved its antiseizure and antiepileptogenic effects in a variety of animal models. Therefore, higher serum BDNF in epilepsy patients, correlated with the severity of disease, may reflect adaptations to seizures. Moreover, inhibiting BDNF-TrkB signaling and simultaneous activation of the NPY system could become a new treatment pathway for epilepsy [45,46]. In fact, some widely recognized antiepileptic drugs may act this way. For instance, valproate was reported to increase the BDNF level and then downregulate BDNF/TrkB protein signaling in the hippocampus [47]. Interesting results were shown in the study of Tegkul et al. [48], where 14-day treatment with subconvulsive doses of pentetrazole decreased the brain level of BDNF. In contrast, BDNF levels were increased when pentetrazole was co-administered with midazolam or levetiracetam [49].

According to experimental data, an intrahippocampal injection of GDNF, as well as the implementation of viral vectors expressing GDNF, can suppress chemically- and electrically-induced seizures [49]. A wide range of research has indicated that chronic treatment with different classes of antidepressant drugs increases brain concentrations of BDNF. In one of the first reports, tranylcypromine, sertraline, desipramine, and mianserin significantly increased BDNF mRNA in the rat hippocampus [50]. Moclobemide increased the BDNF level in in vitro experiments conducted on hippocampal progenitor cells [51]. Fluoxetine increased the hypothalamic BDNF expression in mice [52], while milnacipran enhanced BDNF protein and mRNA levels in the mouse cerebral cortex [53]. On the other hand, duloxetine activated the BDNF protein expression in brains of rats subjected to the methamphetamine model of anxiety [54]. Desipramine, fluoxetine, and tianeptine increased the BDNF mRNA and protein expression in stressed rats [55]. Finally, in clinical conditions, increased serum BDNF levels were detected in 30 patients treated with bupropion [56].

Not only BDNF, but also GDNF, are being considered as potential therapeutic agents in the antiepileptic strategy. Similarly to BDNF, the brain levels of GDNF are elevated by a variety of antidepressants with different mechanisms of action, including amitriptyline, clomipramine, mianserin, fluoxetine, paroxetine [57,58,59], and mirtazapine [60].

Interestingly, escitalopram increased the serum levels of both GDNF and BDNF in obsessive-compulsive disorder in rats [61]. Fluoxetine, imipramine, and milnacipran upregulated BDNF and GDNF in the striatum and substantia nigra of rats in the MPTP model of Parkinson’s disease [62].

Probable mechanisms of the antiseizure action of antidepressant drugs and their influence on the antielectroshock action of antiepileptics are shown in Figure 1.

### 4.2. Considerations Resulting from Analyzed Data

Doses of antidepressant drugs affecting the action of antiepileptics have been converted to human doses according to Raegan-Shaw et al. [63]. Only in the case of reboxetine and tianeptine did the doses used in experiments exceed the maximal single doses of these antidepressants. Therefore, it seems that the likelihood of interactions is relatively high in clinical conditions (Table 3).

In general, most antidepressant drugs, given either acutely or chronically, exhibited their own antiseizure action, increasing the electroconvulsive threshold in mice. The exception was chronic mianserin, which led to a proconvulsive effect in this test. From a theoretical point of view, there is one possible explanation for this—mianserin behaves as an antagonist of the dopamine D_2_ receptor (DrugBank https://go.drugbank.com/drugs/DB06148, accessed on 21 January 2021). The anticonvulsant action of antiepileptic drugs was enhanced by both acute and chronic treatment with antidepressant drugs. Therefore, adaptive changes with monoaminergic receptors necessary to develop an antidepressant effect do not seem to be crucial for their antielectroshock activity. With regards to conventional antiepileptic drugs, their antiseizure action was potentiated by almost all groups of antidepressants. However, fluoxetine, which is a representative of selective serotonin reuptake inhibitors, increased the brain concentration of all antiepileptics tested, including valproate, carbamazepine, phenytoin, and phenobarbital. Therefore, interactions between fluoxetine and classical antiseizure drugs have a strong pharmacokinetic component.

Less pharmacokinetic interactions were found when antiepileptic drugs were combined with representatives of remaining groups of antidepressants (venlafaxine, milnacipran, duloxetine, reboxetine, mianserin, and even moclobemide). The common mechanism of action of these drugs is the enhancement of noradrenergic transmission. Therefore, the question of whether it is not noradrenaline that is responsible for the enhancement of the antiseizure action arises. Nevertheless, the fact that chronic mianserin decreased the action of valproate and phenytoin does not support this hypothesis. It can only be speculated that this effect must be related to the downregulation of 5-HT_2_ and α_2_ receptors.

It is even more difficult to explain why trazodone, increasing the electroconvulsive threshold after chronic treatment, attenuated the antielectroshock action of carbamazepine and phenytoin. However, recent data showed that trazodone enhanced seizures in a genetic WAG/Rij ((Wistar Albino Glaxo from Rijswijk) rat model of absence epilepsy [64]. Perhaps the mechanisms responsible for this proconvulsive action are also involved in interactions with carbamazepine and phenytoin. It remains unclear why just phenytoin is so prone to the attenuating action of mianserin and trazodone. Simultaneously, phenytoin also seemed to be the most susceptible to the potentiating action of remaining antidepressant drugs. Further studies are required to elucidate this issue.

Practically, antidepressant drugs enhanced, to varying degrees, the action of all antiepiletics tested, regardless of their main mechanism of action—sodium channel blockade, enhancement of GABA-ergic, or attenuation of glutamatergic neurotransmission. However, the strongest interactions were observed between acute/chronic tianeptine and phenobarbital, venlafaxine, and phenytoin, as well as between chronic bupropion and topiramate. In these cases, the antielectroshock action of phenobarbital and topiramate was enhanced, despite their lowered brain concentrations. Interestingly, phenobarbital and topiramate share one mechanism of action—the two drugs bind to the GABA receptor subunit alpha-1. Perhaps the influence on this subunit contributes to such a strong response to antidepressant drugs. It is quite surprising, however, that tianeptine, which is an antidepressant increasing the serotonin uptake, potentiates the action of antiepileptics as well as or even better than antidepressant drugs enhancing monoaminergic transmission. This may suggest that factors other than monoaminergic mechanisms are involved in these interactions. The most obvious assumption is that antiepileptic and antidepressant drugs synergistically decrease glutamatergic and/or increase GABA-ergic neurotransmission. The two groups of drugs may also increase the brain BDNF and/or GDNF expression. Both hypotheses require verification in further research.

## 5. Conclusions

Most antidepressant drugs tested exhibited an antiseizure or neutral action in the electroconvulsive threshold test in mice. The exception was chronic mianserin, which displayed a significant proconvulsant effect. Either acute or chronic treatment with antidepressants potentiated or did not affect the antielectroshock action of classical anti-epileptic drugs. Only chronic mianserin and acute and chronic trazodone diminished the action of valproate, phenytoin, and carbamazepine in the MES test. There are no premises that the main mechanisms involved in the antidepressant and/or antiseizure action may be responsible for interactions between the two groups of medications. Therefore, it can be assumed that other factors provide the background for the effect of drug combinations. BDNF and GDNF modulate both antiseizure and antidepressant-like effects in experimental conditions. Their role in these interactions, as very likely, should be confirmed in further investigations.

## Figures and Tables

**Figure 1 ijms-22-02521-f001:**
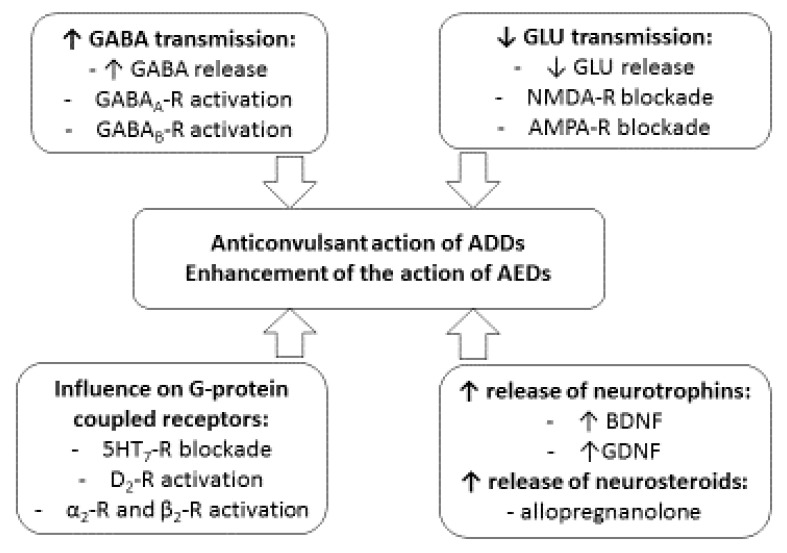
Probable mechanisms of the anticonvulsant action of antidepressant drugs (ADDs) and their effect on the antielectroshock action of antiepileptic drugs (AEDs).

**Table 1 ijms-22-02521-t001:** Effects of antidepressant drugs on the electroconvulsive threshold and the antielectroshock action of antiepileptic drugs.

Antidepressant Drug	Effect on the ECT	Effect on the Action of AEDs	Effect on Brain Levels of AEDs	Reference
Acute amitriptyline	↑	↑ VPA, ↑ CBZ, ↑ PB	nt	[7]
Acute imipramine	↑	↑ VPA	nt	[7]
Acute desipramine	↔	↑ VPA	↔	[7]
Acute fluoxetine	↑	↑ VPA, ↑ CBZ, ↑ PB, ↑ PHT	↑ CBZ, ↑ PB, ↔ PHT	[12]
Chronic fluoxetine	↔	↑ VPA, ↑ CBZ, ↑ PB, ↑ PHT	↑ VPA, ↑ CBZ, ↑ PB, ↑ PHT	[13]
Acute venlafaxine	↑	↑ VPA, ↑ CBZ, ↑ PB, ↔ PHT	↔ VPA, ↔ CBZ, ↑ PB	[17]
Chronic venlafaxine	↑	↑ PHT, ↔ VPA, ↔ CBZ, ↔ PB	↓ PHT	[17]
Acute milnacipran	↑	↑ VPA, ↑ CBZ, ↑ PB, ↑ PHT	↔ VPA, ↔ CBZ, ↔ PB, ↔ PHT	[18]
Chronic milnacipran	↔	↔ VPA, ↔ CBZ, ↔ PB, ↔ PHT	↔ VPA, ↔ CBZ, ↔ PB, ↔ PHT	[18]
Acute duloxetine	↑	↑ VPA, ↑ CBZ, ↑ OXC	nt	[19]
Acute reboxetine	↑	↑ VPA, ↑ CBZ, ↑ PB, ↑ PHT	↔ VPA, ↔ CBZ, ↔ PB, ↔ PHT	[20]
Chronic reboxetine	↔	↑ CBZ, ↔ VPA, ↔ PB, ↔ PHT	↔ CBZ	[20]
Chronic bupropion	↑	↑ FBM, ↑ LTG, ↑ TPM	↔ FBM, ↑ LTG, ↓ TPM	[23]
Acute moclobemide	↑	↑ VPA, ↑ CBZ, ↑ PB, ↔ PHT	↑ VPA, ↑ CBZ, ↔ PB	[24]
Chronic moclobemide	↔	↑ VPA, ↑ CBZ, ↑ PB, ↑ PHT	↑ VPA, ↑ CBZ, ↔ PB, ↑ PHT	[24]
Acute mianserin	↑	↑ VPA, ↑ CBZ, ↑ PHT, ↔ PB	↔ VPA, ↔ CBZ, ↔ PHT	[25]
Chronic mianserin	↓	↓ VPA, ↓ PHT, ↔ CBZ, ↔ PB	↔ VPA, ↔ PHT	[26]
Acute trazodone	↔	↓ CBZ, ↓ PHT,↔ VPA, ↔ PB	↔ CBZ, ↔ PHT	[26]
Chronic trazodone	↑	↓ CBZ, ↓ PHT, ↔ VPA, ↔ PB	↔ CBZ, ↔ PHT	[26]
Acute tianeptine	↔	↑ VPA, ↑ CBZ, ↑ PB, ↔ PHT	↔ VPA, ↔ CBZ, ↓ PB	[28]
Chronic tianeptine	↔	↑ VPA, ↑ CBZ, ↑ PB, ↔ PHT	↔ VPA, ↔ CBZ, ↓ PB	[28]

Acute treatment means that a given antidepressant drug was injected only once, while chronic treatment means that this drug was applied for 14 days. Brain concentrations of antiepileptic drugs were only measured in the case of positive antielectroshock interactions. AEDs, antiepileptic drugs; CBZ, carbamazepine; ECT, the electroconvulsive threshold; FBM, felbamate; LTG, lamotrigine; nt, not tested; PB, phenobarbital; PHT, phenytoin; TPM, topiramate; VPA, valproate; ↑, increased electroconvulsive threshold/potentiated antielectroshock action/increased brain concentration; ↓, decreased electroconvulsive threshold/reduced antielectroshock action/decreased brain concentration; ↔, no effect on the aforementioned parameters.

**Table 2 ijms-22-02521-t002:** Effects of antidepressant drugs on the antielectroshock action of individual antiepileptic drugs.

AEDs	ADDs Enhancing the Action of a Given AED	ADDs Attenuating the Action of a Given AED
Valproate	aAMI, aIMI, aDES, chrFXT, aVLF, aMLN, aDUL	na
	aRBX, aMNS, aTNP, aMCB, chrMCB	
Carbamazepine	aAMI, aFXT, chrFXT, aVLF, aMLN, aDUL, aRBX	chrMNS, aTZD, chrTZD
	chrRBX, aMNS, aTNP, chrTNP, aMCB, chrMCB	
Phenobarbital	aAMI, aFXT, chrFXT, aVLF, aMLN, aRBX, aTNP	na
	chrRBX, aMNS, aTNP, chrTNP, aMCB, chrMCB	
Phenytoin	aFXT, chrFXT, chrVLF, aMLN, aRBX, aMNS, chrMCB	chrMNS, aTZD, chrTZD
Felbamate	chrBUP	na
Lamotrigine	chrBUP	na
Topiramate	chrBUP	na

a, acute treatment; ADDs, antidepressant drugs; AEDs, antiepileptic drugs; AMI, amitriptyline; BUP, bupropion; chr, chronic treatment; DES, desipramine; DUL, duloxetine; FXT, fluoxetine; IMI, imipramine; MCB, moclobemide; MLN, milnacipran; MNS, mianserin; na, not applicable; RBX, reboxetine; TZD, trazodone.

**Table 3 ijms-22-02521-t003:** Conversion of minimal doses of antidepressant drugs affecting the action of antiepileptic drugs against MES in mice to human doses.

Antidepressant Drug	MD (mg/kg)	cHD (mg/kg)	cHD/70 (mg)	Maximal Single HTD (mg)	DI
Amitriptyline	10	0.81	56.7	100	1.76
Fluoxetine	10	0.81	56.7	60	1.0
Venlafaxine	10	0.81	56.7	150	2.65
Milnacipran	5	0.40	28.0	50	1.76
Duloxetine	6.25	0.51	35.7	120	3.36
Reboxetine	4	0.32	22.4	12	0.56
Bupropion	5	0.81	56.7	150	2.65
Moclobemid	50	4.05	283.5	300	1.06
Mianserin	10	0.81	56.7	200	3.53
Trazodone	5	0.40	28.0	300	10.71
Tianeptine	50	4.05	283.5	12.5	0.04

MD, mouse dose; cHD, converted human dose; cHD/70, converted human dose per patient weighing 70 kg; HTD, human therapeutic dose; DI, dose index—the cHD/70 to maximal single HTD ratio.

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
