# Peer review of "How Antidepressant Drugs Affect the Antielectroshock Action of Antiseizure Drugs in Mice: A Critical Review"

_ijms, 2021, doi:10.3390/ijms22052521_

Round 1

Reviewer 1 Report

Borowicz-Reutt reviewed the current literature on the interactions of antidepressants with antiepileptic drugs in maximal electroshock (MES) in mice. The review is interesting; however, some things could be improved:

    1. Introduction: "The coexistence of these disease 26 entities reaches 30% and depressive episodes begin to be considered as a marker of drug- 27 resistant epilepsy." needs a reference.
    2. I suggest updating the manuscript regarding the newer antidepressants, i.e., ketamine (or esketamine), brexanolone, vortioxetine, or agomelatine, and their effects in MES. If no studies could be found, this should also be mentioned in the manuscript.
    3. Table 1: The table is not clear enough. What column title "Pharmacokinetics" means? I would suggest changing the column title. What is ECT? The acronyms should be explained in alphabetical order.
    4. Table 2: I understand that "a" is acute and "chr" is chronic, but this should be explained in the table legend. What arrows mean in this table? Similar to Table 1, Table 2 is difficult to understand
    5. The manuscript would benefit from a more thorough edit to improve the awkward phrasing and readability.

Author Response

Responses to the Reviewer’s 1 comments

I would like to express my thanks for all the comments raised and hope that the amended manuscript will fulfill all expectations. For the convenience of the reviewers, "Track Changes" function was used to visualize corrections. I realize that corrections  may occur not satisfactory. If so, I would like to kindly ask for more detailed directions on how to improve the manuscript.

    1. Introduction: "The coexistence of these disease 26 entities reaches 30% and depressive episodes begin to be considered as a marker of drug- 27 resistant epilepsy." needs a reference.

In the end of the 1st para of the Introduction the respective reference has been added (Pg. 1).

    1. I suggest updating the manuscript regarding the newer antidepressants, i.e., ketamine (or esketamine), brexanolone, vortioxetine, or agomelatine, and their effects in MES. If no studies could be found, this should also be mentioned in the manuscript.

As requested by the Reviewer, interactions between ketamine and conventional antiepileptics were described in the new subchapter: 3.8. Newer drugs with antidepressant activity. Indeed, no literature data could be found on the action of esketamine, brexanolone, vortioxetine, or agomelatine in the MES test. However, the effect of brexanolone (allopregnanolone) on the action of tiagabine was studied in two other test based on electrically-evoked seizures. The respective fragment can be found on Pgs. 4-5.

In the 3rd para of the Discussion, I have added a sentence broadening information about possible antielectroshock mechanisms of ketamine (Pg. 5).

    1. Table 1: The table is not clear enough. What column title "Pharmacokinetics" means? I would suggest changing the column title. What is ECT? The acronyms should be explained in alphabetical order.
    2. Table 2: I understand that "a" is acute and "chr" is chronic, but this should be explained in the table legend. What arrows mean in this table? Similar to Table 1, Table 2 is difficult to understand

All suggested corrections were made in Tables. The word “Pharmacokinetics” was replaced with “Effect on brain levels of AEDs”. The Table 2 was re-arranged to increase its readability.  Results were placed in two columns and arrows were entirely eliminated.

    1. The manuscript would benefit from a more thorough edit to improve the awkward phrasing and readability.

            The manuscript has been corrected linguistically by a professional translator.

Reviewer 2 Report

This is an interesting review summarizing the current data on the effect of antidepressant drugs on the activity of antiseizure drugs in the maximal electroshock seizure test. The paper is well-written and very informative.

The only consideration I would like to make is the lack of scheme(s)/figure(s) on the possible mechanism underlying the pharmacodynamic interactions between some antidepressant and antiseizure drugs.

Author Response

The only consideration I would like to make is the lack of scheme(s)/figure(s) on the possible mechanism underlying the pharmacodynamic interactions between some antidepressant and antiseizure drugs.

I would like to thank the Reviewer for this suggestion. As it was requested, a figure describing antielectroshock mechanisms of antidepressant drugs has been added. I hope it is informative enough to meet expectation of the Reviewer. If not, I would kindly ask for more suggestions on how to improve this figure.